# Diclofenac Embedded in Silk Fibroin Fibers as a Drug Delivery System

**DOI:** 10.3390/ma13163580

**Published:** 2020-08-13

**Authors:** Alena Opálková Šišková, Erika Kozma, Andrej Opálek, Zuzana Kroneková, Angela Kleinová, Štefan Nagy, Juraj Kronek, Joanna Rydz, Anita Eckstein Andicsová

**Affiliations:** 1Polymer Institute, Slovak Academy of Sciences, Dúbravská cesta 9, 84541 Bratislava, Slovakia; alena.siskova@savba.sk (A.O.Š.); zuzana.kronekova@savba.sk (Z.K.); angela.kleinova@savba.sk (A.K.); juraj.kronek@savba.sk (J.K.); 2Institute for Chemical Sciences and Technologies ‘Giulio Natta’ (SCITEC-CNR), Via A. Corti 12, 20133 Milan, Italy; erika.kozma@scitec.cnr.it; 3Institute of Materials and Machine Mechanics, Slovak Academy of Sciences, Dúbravská cesta 9, 84541 Bratislava, Slovakia; andrej.opalek@savba.sk (A.O.); nagy.stefan@savba.sk (Š.N.); 4Centre of Polymer and Carbon Materials, Polish Academy of Sciences, 34, M. Curie-Skłodowska St., 41-800 Zabrze, Poland; jrydz@cmpw-pan.edu.pl

**Keywords:** silk fibroin, nanofibers, electrospinning, casein, diclofenac, drug delivery system

## Abstract

Silk fibroin is a biocompatible, non-toxic, mechanically robust protein, and it is commonly used and studied as a material for biomedical applications. Silk fibroin also gained particular interest as a drug carrier vehicle, and numerous silk formats have been investigated for this purpose. Herein, we have prepared electrospun nanofibers from pure silk fibroin and blended silk fibroin/casein, followed by the incorporation of an anti-inflammatory drug, diclofenac. Casein serves as an excipient in pharmaceutical products and has a positive effect on the gradual release of drugs. The characteristics of the investigated composites were estimated by scanning electron microscope, transmission electron microscope, thermogravimetric analysis, and a lifetime of diclofenac by electron paramagnetic resonance analysis. The cumulative release in vitro of diclofenac sodium salt, together with the antiproliferative effect of diclofenac sodium salt-loaded silk nanofibers against the growth of two cancer cell lines, are presented and discussed.

## 1. Introduction

During the last decade, both synthetic and natural drug-release materials have been developed for use in drug delivery systems. Most of the drug delivery formulations on the market or in research are based on synthetic polymers, such as polyesters, polyphosphazene, or poly(lactic-*co*-glycolic acid), due to their desirable pharmacokinetics and controllable hydrolytic degradation profiles [1]. Although they are generally considered safe, their inherent properties and processing requirements limit their use in certain sustained delivery areas, such as protein therapeutics, where these issues can impact product stability.

Naturally derived biopolymers, such as gelatin, alginates, collagen and cellulose, offer an attractive alternative to synthetic polymers and are currently under investigation for their use in drug delivery formulations [2]. Since most of these polymers tend to have relatively high dissolution rates in aqueous media, batch-to-batch variability and sourcing-related issues, the development of protein-based formulation must offer tunable sustained release kinetics and enhanced product stability from a reliable source and a well-characterized starting material. To fulfill these requirements, a considerable amount of work has been conducted in the area of silk protein-based materials for drug delivery applications [3].

Silk fibroin (*Bombyx mori*) is a biologically derived protein that meets the biomaterial requirements for wound healing, implantable, and degradable medical applications [4,5]. There are already well-established products on the market, such as silk sutures (Surusil^®^, Sofsilk™), silk scaffolds (Seri^®^) and bone tissue engineering LAPONITE^®^ [6].

Silk fibroin presents a combination of beneficial properties for drug delivery, including excellent biocompatibility [7], degradation to non-toxic products in vivo [8]. It provides easily accessible functional groups for chemical modifications [9].

Various silk-based material formats have been investigated as drug carriers, including bulk loaded systems, microspheres, nanoparticles or nanofibers [10,11,12]. Nanofibers with nanoscale structures are generated by electrospinning when high voltage is applied to the polymer solution. The formed nanofibers are characterized by a large surface area, high porosity, and superior mechanical properties [13,14,15,16]. The silk fibroin-based fibers (SFs) can be obtained either from pure silk fibroin or blended with other materials such as synthetic or natural polymers or small molecules. The presence of other components in SFs can influence the silk physical properties or can enrich their functionalities [17,18,19,20]. A suitable candidate that can modify the drug release properties due to its structural and physicochemical properties is casein. This major milk protein is also inexpensive, non-toxic, readily available, and stable [21]. Casein subunits are amphiphilic, and it has been suggested that casein acts as a stabilizer of protein structure [22]. The use of casein for drug delivery purposes in various formulations generates a slow release of the active materials [23,24,25]. 

The advantages of drug delivery from fibers include the short diffusion path of small molecules from the biomaterial to the issue, which will yield a higher overall release rate than the corresponding bulk material. The release of the drugs at a defined rate over a definite period should be possible by tuning polymer properties such as polymer structure, nanofibers morphology, composition, orientation of the fibers, and fibrous web porosity [26,27].

The active component for the silk fibroin nanofibers and casein-mixed silk fibroin nanofiber (CSFs) delivery system is one of the most commonly used drugs in the world. Diclofenac, a non-steroidal anti-inflammatory drug, used in the treatment of pain in rheumatoid arthritis, migraine, and post-operative pain, has an established role in oncological practice in the treatment of cancer-related pain and as a topical treatment for actinic keratosis, which is commonly viewed as a pre-cancerous lesion. The treatment of various types of cancer cell lines like fibrosarcoma [28], colorectal cancer [29], neuroblastoma [30], and ovarian cancer [31,32] showed a reduced growth rate and low levels of vascularization.

The main objective of this study was to prepare biobased composite materials and to use them as delivery systems for diclofenac, with sustained releasing kinetics. Accordingly, in this work, we describe the preparation of electrospun SFs and CSFs as drug delivery carrier vehicles for the in vitro release of diclofenac sodium salt as a model system. Then, we have also studied the antiproliferative activity of diclofenac sodium salt-embedded electrospun silk mats against cancer human cells derived from the skin (CaSki, epidermoid carcinoma derived from the metastatic site in the small intestine) and cervix (Hela cells).

## 2. Materials and Methods

### 2.1. Materials

Commercially degummed silk fiber from *Bombyx mori* was used for our experiments. Casein (Merck, Darmstadt, Germany), lithium bromide (Alfa Aesar, Karlsruhe, Germany), ethanol (Centralchem, Bratislava, Slovakia), toluene (Microchem, Pezinok, Slovakia), 1,1,1,3,3,3-hexafluoro-2-propanol (HFIP, hexafluoroisopropanol, TCI Tokyo Kasei, Japan), diclofenac sodium salt (DSS) and dimethyl sulfoxide (Sigma-Aldrich, Weinheim, Germany), (2,2,6,6-tetramethylpiperidin-1-yl)oxyl (TEMPO•, Sigma-Aldrich, Poznań, Poland) were used without further purification. Methanol and water for high-performance liquid chromatography (HPLC) analysis were purchased as HPLC grade solvents from Macron (Gliwice, Poland) and CentralChem (Bratislava, Slovakia), respectively. 3-(4,5-Dimethyldiazol-2-yl)-2,5-diphenyltetrazolium bromide (MTT) was purchased from Calbiochem (Merck Millipore, Darmstadt, Germany). Dulbecco’s modified Eagle medium (DMEM), fetal calf serum (FCS), streptomycin, penicillin, and l-glutamine were purchased from Gibco (Life Technologies, Grand Island, NY, USA). Human cell lines: MRC5 normal lung fibroblasts, CaSki epidermoid carcinoma derived from the metastatic site in the small intestine, and Hela cervix carcinoma were obtained from the collection in the Institute of Virology BMC SAV (Bratislava, Slovakia).

### 2.2. Processing of Natural Silk

A total of 2 g of silk was dissolved in 10 mL of a 9.3 M lithium bromide solution by stirring at 60 °C for 4 h to obtain a 20% *w*/*v* solution. The homogeneous solution was dialyzed in distilled water using the dialysis membrane (12–14 kDa, Sigma-Aldrich, Saint Louis, MO, USA) for 2 days. The aggregates that occurred during dialysis were removed by centrifugation (10 min, 10,000 rpm, 25 °C). The final concentration of the aqueous silk solution was approximately 8% *w*/*v*, which was subsequently lyophilized and then dissolved in HFIP for electrospinning.

### 2.3. Preparation of Solutions for Electrospinning

The components for solutions used in the electrospinning are listed in Table 1.

For all compositions, casein content was selected to be 5%, which is the upper limit of its solubility in our experimental conditions.

### 2.4. Electrospinning

Fiber mats were prepared using the electrospinning method. This process was carried out under ambient temperature in a horizontal spinning configuration. The prepared solutions were placed into a 10 mL syringe bearing a flat-end needle with a 0.8 mm inner diameter. The needle was connected with a high voltage power supply (Spellman SL-150W, Bochum, Germany). The applied voltage was held at 20 kV with positive polarity. Applied voltages were selected on the basis of results from previous unpublished experiments with the individual solutions. The working distance (distance between the top of the needle and collector) was 10 cm. The solutions were fed by an NE-1000 (New Era Pump Systems, Inc., Farmingdale, NY, USA) single syringe pump model with a constant volume rate of 1 mL·h^−1^. The electrospun fibers were collected on 15 × 15 cm aluminum foil.

### 2.5. Characterization

#### 2.5.1. Scanning Electron Microscopy (SEM)

The morphology of electrospun silk fibers and the average diameter of the fibers were determined by a SEM JEOL JSM-6610 (Tokyo, Japan) microscope at an accelerated voltage of 10 kV. The samples were sputtered with a thin layer of gold. AzTec (Springfield, NJ, USA) software was used for collecting SEM images and for processing the results. The images were post-processed using Image J software (LOCI, University of Wisconsin, Madison, WI, USA). Approximately 50 fiber segments were analyzed randomly to obtain a mean diameter for each nonwoven fiber.

#### 2.5.2. Transmission Electron Microscopy (TEM)

Transmission electron microscope JEOL 1200FX (Tokyo, Japan) operated at 80 kV was used. Samples were placed and observed on the copper grids.

#### 2.5.3. Attenuated Total Reflectance–Fourier Transform Infrared Spectroscopy (ATR-FTIR)

ATR-FTIR spectra were recorded on a spectrophotometer Nicolet 8700 (Thermo Fisher Scientific, Madison, WI, USA) using KBr, with a deuterated triglycine sulfate and thermoelectricity cooled (DTGS TEC) detector in the region 4000–600 cm^−1^ at a resolution of 4 cm^−1^ using the absorbance mode.

#### 2.5.4. Elemental Analysis

Elemental analysis (C, H, N, S) was measured using the FLASH 2000 organic elemental analyzer (Thermo Fisher Scientific, Waltham, MA, USA).

#### 2.5.5. Thermal Analysis

Thermogravimetric analysis (TGA) was performed on a Mettler Toledo (Columbus, OH, USA) thermal analyzer in the temperature range of 24–500 °C with a heating rate of 10 °C/min in a stream of nitrogen, and the nitrogen gas flow rate was 100 mL/min. Approximately 1–3 mg of each sample was weighed, sealed in an aluminum pan and measured with an empty pan as a reference.

#### 2.5.6. Contact Angle Measurements

Surface wettability of the free-standing electrospun scaffolds with thickness 25 ± 1 μm was characterized by the water contact angle measurements. Static contact angle measurements of all electrospun mats were performed at room temperature (22 ± 1 °C). At the analysis, 10 µL droplet of distilled water as the reference liquid was deposited by syringe pointed vertically down onto the sample surface. A Canon Power Shot SX130 (Tokyo, Japan) camera was used for taking images. The photography was taken in the moment that the water droplet hit the surface. The results were processed by Image J software. The contact angle was measured five times (n = 5) from different positions, and an average value was calculated by the statistical method.

#### 2.5.7. Electron Paramagnetic Resonance (EPR) Analysis

The EPR analysis was made using Bruker WinEPR Processing (Billerica, MA, USA). Blank solution (50 μM) was prepared from 0.78 mg of TEMPO• in 10 mL HFIP. The concentration of the investigated DSS was 1.48 mg/mL. The EPR signals were recorded 5 min after the start of the reaction under the following conditions: field modulation of 100 KHz, modulation amplitude of 1 G, field constant of 40.96 ms, conversion time of 671.089 ms, center field of 3,245 G, sweep width of 100 G, X-band frequency of 9.64 GHz, power of 20 mW and temperature of 25 °C.

#### 2.5.8. DSS Release from Fibrous Nanostructures

The DSS release was performed from electrospun mats containing 15% *w*/*w* of DSS and 5% *w*/*v* of casein. DSS-loaded silk mats of defined geometry (1 cm in diameter, < 1 mm in thickness and mass of approx. 2 mg) were put into 1.5 mL phosphate buffer solution (PBS, pH 7.4) at 37 °C.

Aliquots of PBS after 24 and 48 h incubation with DSS-loaded mats were taken and analyzed by HPLC. HPLC was conducted in the reverse phase using an Agilent 1200 equipment (Agilent, Santa Clara, CA, USA) equipped with an isocratic pump G1310A, UV/Vis detector (VWD G1314B), column ZORBAX Eclipse Plus C18 150 × 4.6 mm and a manual injector of Rheodyne 7725i type with a loop volume of 20 μL. All measurements were performed in methanol/water (90:10) used as the mobile phase in ambient temperature, with a flow rate of 0.5 mL/min and a detection wavelength of 280 nm. All measurements were conducted in duplicates for three independent samples (n = 6). In all release measurements, the sample for HPLC was prepared by diluting 10 µL of PBS solution of released DSS with 90 µL methanol. The amount of released DSS was estimated from the calibration curve. The calibration curve for DSS concentration was conducted in methanol:water (90:10), methanol:PBS (90:10), and methanol:DMEM (90:10) using the same conditions of measurements as in the case of extracts from silk mats. For all solvents, the signal for DSS was not overlapped with residual signals, and all points can be fitted with the same calibration curve.

#### 2.5.9. Cell Viability Assay

Empty and DSS-loaded silk casein mats of defined geometry (1 cm in diameter and ~1 mm thickness) were prepared by electrospinning as described above, and sterilized for 30 min using a germicidal lamp GT20T10, 20w, 254 nm (Sankyo Denki, Tokyo, Japan). Cells were seeded in 12-well tissue culture plates at concentrations of 1 × 10^5^ per well and incubated overnight in full growth medium (DMEM, 10% FCS, penicillin and streptomycin) in a CO_2_ incubator at 37 °C, 5% CO_2_ and saturated humidity. At the start of experiment, 1.5 mL of culture media was added into cells and one silk mat was placed in the well. Incubation and extraction of DSS were performed at 24 and 48 h, respectively. All samples were conducted in quadruplicates (n = 4).

#### 2.5.10. MTT Assay

Cell viability was assessed by using standard colorimetric MTT assay. After 24 or 48 h of cell treatment, the empty and DSS-loaded silk mats were removed from treated cells and culture medium was replaced with fresh culture medium containing 0.5 mg/mL MTT and incubated 2 h. After incubation, the medium was removed and water-insoluble purple formazan as a product of mitochondrial metabolism in living cells was dissolved in dimethylsulfoxide (DMSO). The absorbance of the DMSO extract was measured using a Multiskan™ FC Microplate Photometer (Thermo Fisher Scientific, Waltham, MA, USA) at 595 nm.

#### 2.5.11. Statistical Data Analysis

Experiments were performed at least three times. The average diameters of the fibers were estimated statistically from at least 50 measured values. Data were expressed as the mean ± standard deviation (SD). For cell experiments, all results are presented as mean ± SD from quadruplicates (n = 4) or as mean ± SEM from three independent experiments. Statistical analysis was performed using one way ANOVA and subsequently the Tukey test of significance.

## 3. Results and Discussion

### 3.1. Silk Nanofibers Morphology

In the present work, randomly oriented, smooth and continuous bead-free silk composite nanofibers were produced by electrospinning, with structure-loaded 15% and 30% *w*/*w* diclofenac sodium salt and/or 5% *w*/*v* casein (see Table 1). The morphology of the fibrous composites is shown on SEM micrographs in Figure 1.

Hexafluoroisopropanol was selected as an appropriate solvent for electrospinning since all components (silk, casein and DSS) are readily soluble in our experimental conditions. Moreover, it was demonstrated that the HFIP-prepared silk scaffolds are more stable than aqueous-based silk composites [33]. The most suitable silk fibroin concentration for fiber formation, optimized by gradual experiments, was determined to be 8% *w*/*v*. At the same time, other parameters, such as voltage, flow rate, and distance, were held constant (see Materials and Methods). The average fiber’s diameters and fiber diameter distributions were calculated from the SEM micrographs, as shown in Figure 1. The sizes of electrospun silk fibroin composite fibers with various compositions are summarized in Table 2.

The fiber average diameter of pure silk fibroin is 162 ± 97 nm (Table 2, Figure 1a). The addition of DSS at a concentration of 15% *w*/*w* had no influence on the fiber diameter, having the same value as the pure electrospun SFs (Figure 1a,c). Only the standard deviation was increased. When the concentration of DSS changed from 15% to 30% *w*/*w*, the fiber diameters grew from 185 nm to 238 nm (Figure 1c,d). The increase in standard deviation in both concentrations indicates less uniform fibers in comparison to the pure silk nanofibers. The presence of polar DSS also had a direct impact on the hydrophilicity of the silk composite scaffolds. Indeed, the contact angles of the SF/DSS composites had lower values in comparison with the pure silk scaffolds, indicating a higher hydrophilicity.

By adding 5% *w*/*w* casein to the silk fibroin solution, the fiber average diameter increased to 301 ± 289 nm, confirming a good dispersion of casein in SFs (Figure 1b). The effect of casein addition can be determined by measuring the water contact angle. In the case of a pure casein cast film, the contact angle was around 69°. Although the fiber diameters of CSFs were slightly higher when compared to pure SFs, the contact angle of about 69° may indicate the presence of casein on the silk fibroin surface.

With the addition of DSS to the CSFs, the fiber average diameter slightly decreased from 301 ± 289 nm (sample b) to 1993 ± 117 nm (sample e) and 191 ± 137 nm (sample f) for 15% and 30% *w*/*w* DSS, respectively. This effect may have been caused by the conductivity increase in the solution containing DSS. Moreover, passing from 15% DSS to 30% DSS in the CSF composites, the contact angles are somewhat similar, approaching the value of pure silk fibroin scaffolds.

The internal structure of the electrospun 8% *w*/*v* silk nanofibers loaded with 5% *w*/*w* casein and 15% *w*/*w* DSS were also analyzed using TEM. As shown in Figure 2, there were some black outlined objects with diameters about 20 nm in the nanofibers seen from the TEM images. These results confirmed the heterogeneity of the fibrous structure, which indicating that the 5% *w*/*w* casein and 15% *w*/*w* DSS were successfully loaded into the nanofibers.

### 3.2. ATR-FTIR Analysis of SF-Based Composites

Due to the presence of amide groups in silk protein, the characteristic vibration bands at 1620–1655 cm^−1^ are assigned to the absorption peak of amide I (C=O stretching), the bands at 1515–1550 cm^−1^ to amide II (N-H bending) and the bands around 1230–1250 cm^−1^ to amide III (C-N stretching). The molecular conformation of silk fibroin is characterized by *β*-sheet absorption peaks at around 1620, 1515, and 1250 cm^−1^, random coil conformation absorptions at 1640–1650, 1550, and 1240 cm^−1^ as well as *α*-helix absorption at 1655 cm^−1^.

It is well known that protein molecules adopt their native conformation in solution only under specific environmental conditions. Studying the structural and the thermodynamic response of proteins in dependence of solvent conditions, it was found that among different solvents, alcohols, and particularly their fluorinated derivatives, disrupt the native tertiary structure by weakening hydrophobic interactions. With the use of hexafluoroisopropanol as a solvent for the electrospinning process, random coil structures rather than *β*-sheet conformations are expected. Moreover, during the electrospinning process, the evaporation of the solvent is very fast and, therefore, the crystallization of the silk fibroin chains is unlikely. Indeed, the ATR-FTIR spectrum of the as-spun silk fibroin fibers show typical bands at 1652 cm^−1^ (amide I), 1528 cm^−1^ (amide II), and 1230 cm^−1^ (amide III), attributed to the random coil conformation of the silk fibroin (Figure 3).

ATR-FTIR spectra of the CSFs and CSF/DSS composites have small variations in comparison with the pure silk and show absorption peaks at 1651 cm^−1^, 1539–1549 cm^−1^, 1512 cm^−1^ and 1240 cm^−1^, which are characteristic of the same random coil conformation. The ATR-FTIR spectra of the CSFs show an additional absorption band at 1512 cm^−1^, indicating the presence of casein.

### 3.3. Thermal Analysis of SF-Based Composites

The thermal behavior of the electrospun SFs, CSFs, and CSF/DSS composites is depicted in Figure 4. The TGAcurve of the electrospun pure silk shows stability up to 180 °C and the thermal decomposition temperature at 250 °C. TGA curves of electrospun silk containing casein and DSS show two steps of mass loss.

The first mass loss for all fibers of about 8% occurred at temperature between 100 °C and 200 °C due to the loss of water. The second mass loss between 250 °C and 320 °C is more significant and corresponds to the degradation of silk. The addition of casein and diclofenac sodium salt has no significant impact on the thermal characteristic of the fibrous composites.

### 3.4. Electron Paramagnetic Resonance Analysis of Releasing DSS

Electron paramagnetic resonance spectroscopy was used as a powerful technique to study the radical scavenging activity of diclofenac after incorporation of electrospun mats using high voltage. After removing the drug from the CSFs, electrospun mat free radicals were detected using the stable free radical (2,2,6,6-tetramethylpiperidin-1-yl)oxyl. The TEMPO• was reduced when it reacted with the diclofenac donating hydrogen, and this reduction was recorded based on the corresponding inhibition of the EPR spectrum [34]. Here, no significant changes were detected in the quartet resonance of TEMPO•, thus confirming the antioxidant activity of DSS even after release from electrospun mats (Figure 5).

### 3.5. Biocompatibility of Silk Casein Mats and the Antiproliferative Effect of DSS-Loaded Silk Mats

Silk-based composites prepared from silkworm *Bombyx Mori* cocoons have already been used for sutures, and silk–fibroin-based composites have biocompatibility similar to commonly used biomaterials.

To determine the biocompatibility of the silk casein fibroin mats, the cytotoxicity was evaluated using three cell lines: normal fibroblast (MRC5) and two cancer cells (Hela and CaSki). As seen from Figure 6a, the cell viability of healthy human and cancer cells treated with CSF mats was comparable to the cell viability of the untreated cells. The morphology of the cells treated with CSF mats was normal (Figure 6).

Once the cell viability tests were adequate, the diclofenac sodium salt-embedded silk fibroin nanofibers were prepared to investigate the antiproliferative effect. The DSS-embedded electrospun SF mats were used to test the impact on the growth of two cancer cell lines, Hela and CaSki (Figure 6b). The LC_50_ (50% lethal concentration) for Hela and CaSki cells, after 24 h of DSS treatment, were determined to be around 500 µM; therefore, the amount of DSS used in our experiments was 15% *w*/*w*.

The effect of released DSS from silk mats on cell viability was evaluated after 24 and 48 h (Figure 6b) and compared to untreated cells that were considered as 100% for cell viability. A non-significant cytotoxic effect of DSS was observed (82% after 24 h and 79% after 48 h) against MRC5 fibroblasts compared with the untreated cell line (100%). The growth of Hela and CaSki cell lines in the presence of DSS was relatively inhibited after 24 h (76% and 71% for Hela and CaSki cell lines, respectively) when compared to normal MRC5 fibroblasts (84%). The growth of both cancer cell lines was significantly inhibited after 48 h of treatment with DSS-embedded mats when the cell viability decreased to 43% and 47% for Hela and CaSki cell lines, respectively, and remained at 78% for normal fibroblasts (MRC5). The anticancer effect of DSS can be also shown on microscopy images, where in the presence of DSS, signs of nucleus lysis and dead cells among the swollen, round-shaped necrotic cells were observed (Figure 7).

The high voltage applied, which is required for fiber production, had only little or even no influence on drug activity, which is in good agreement with the published literature [35,36,37]. The vast surface area of the electrospun fibers allows efficient and also fast solvent evaporation. Due to this fact, the loaded drug will tend to remain active and will have limited time to recrystallize [38].

The SF-embedded diclofenac nanofibers exhibited a burst phase in the first hour. The release characteristics showed a high percentage of the drug released in the burst phase, followed by a much lower release over the subsequent 48 h. The inclusion of casein was found to significantly modify the release of the drug from silk fibroin mats, with a substantial decrease in the burst phase and an overall release increase to 120 h.

Although the fiber diameters in the CSF composites were higher (231–233 nm) when compared to the pure silk fibroin (172 nm), meaning a larger surface and a more reliable contact with the dissolution medium, a slower release of the drug was observed. Thus, the single component SF nanofiber gave rise to a more significant initial burst and an overall short release. In comparison, the bicomponent CSF composite resulted in a lower burst and higher total release. These results show that by combining two natural polymers, like silk fibroin and casein, it is possible to tune the diclofenac-releasing profile.

In our experimental conditions, by increasing the DSS content from 15% to 30%, both the release time and the relative release percentage remain mostly the same (Figure 8).

## 4. Conclusions

By using electrospinning, electrospun silk fibroin and silk fibroin/casein composites embedded with diclofenac sodium salt were prepared. The composites were tested for cytotoxicity and biocompatibility with fibroblast cells. Moreover, the activity of the drug when released from composites was investigated against cancer human cells derived from the skin (CaSki, epidermoid carcinoma derived from the metastatic site in the small intestine) and cervix (Hela cells).

At the beginning, we revealed that the DSS release mechanism had a burst profile for the pure silk nanofibers and a more sustained release for the CSF composites. Secondly, in this study, we revealed that the tested fibroblasts cells were very well proliferated on SF/DSS and electrospun CSF mats. The mentioned proliferation activity of fibroblast cells indicates that electrospun composite fibers are non-toxic and biocompatible. In comparison to the fibroblast cells, the growth of both cancer lines was significantly inhibited after 48 h. This demonstrated the release of DSS.

Our results indicate that silk fibroin nanofibers are feasible materials for DSS rapid release and that by combining the silk fibroin with casein, a more sustained release can be obtained. Such material could be used as a wound dressing alternative or a drug release system.

## Figures and Tables

**Figure 1 materials-13-03580-f001:**
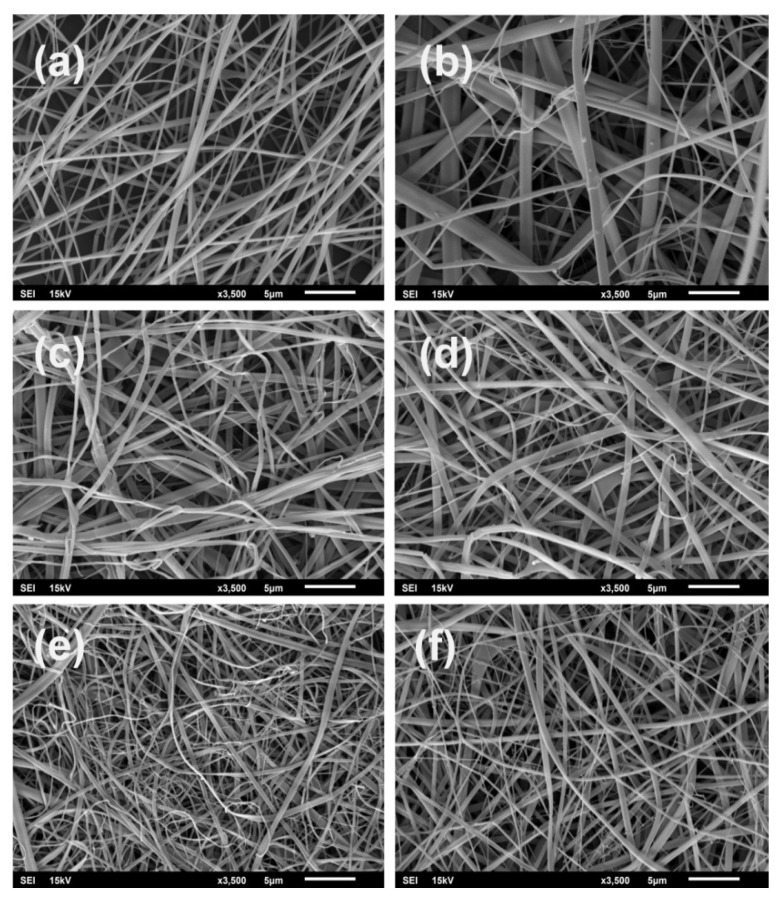
SEM images of electrospun silk fibroin composites: (**a**): SFs, (**b**): CSFs, (**c**): SFs/15%DSS, (**d**): SFs/30% DSS, (**e**): CSFs/15%DSS, (**f**): CSFs/30%DSS.

**Figure 3 materials-13-03580-f003:**
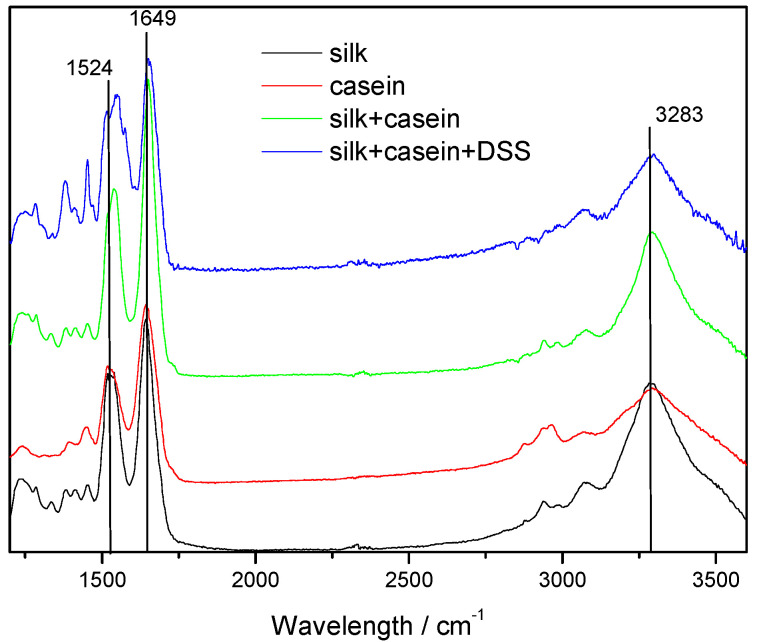
ATR-FTIR spectrum of electrospun nanofibers: silk (SFs, black line), casein (cast film, red line), CSFs (green line), and CSFs/15% DSS (blue line).

**Figure 4 materials-13-03580-f004:**
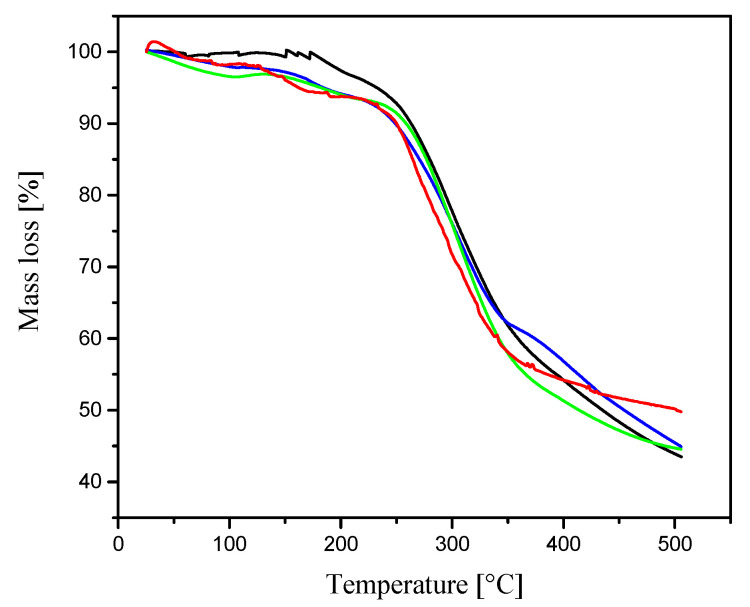
TGA thermograms of electrospun nanofibers: SFs (black line), CSFs (blue line), CSFs/15%DSS (green line), and CSFs/30%DSS (red line).

**Figure 5 materials-13-03580-f005:**
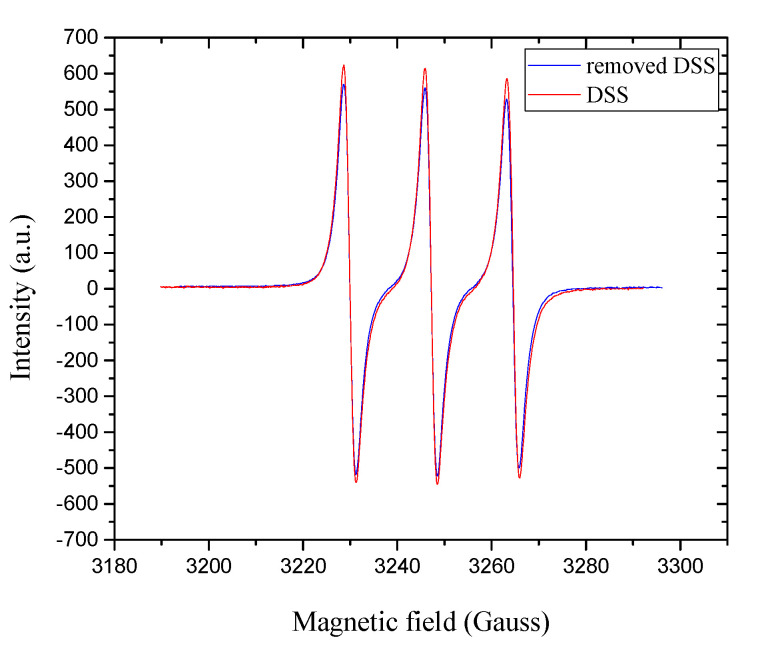
EPR spectra of removed DSS from CSFs/30%DSS (blue line) and the commercial DSS (red line) in HFIP solution in the presence of TEMPO•.

**Figure 6 materials-13-03580-f006:**
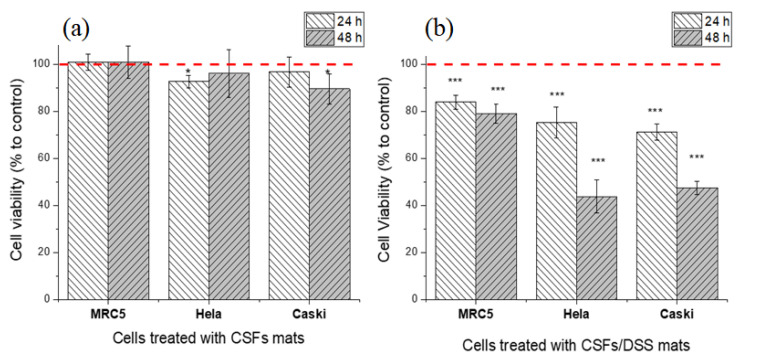
Biocompatibility of CSFs (**a**) and the effect of DSS-loaded CSFs on normal fibroblast (MRC5), Hela and CaSki cells after 24 and 48 h incubation (**b**). The red dashed line corresponds to cell viability of control–untreated cells. Cell viability was determined by MTT assay. The results for the biocompatibility of CSFs (**a**) are presented as mean ± SD (n = 4) and for anti-cancer activity (**b**) as mean ± SEM of three independent experiments. The statistical significance is shown by asterisks (* *p* < 0.05, *** *p* < 0.001).

**Figure 7 materials-13-03580-f007:**
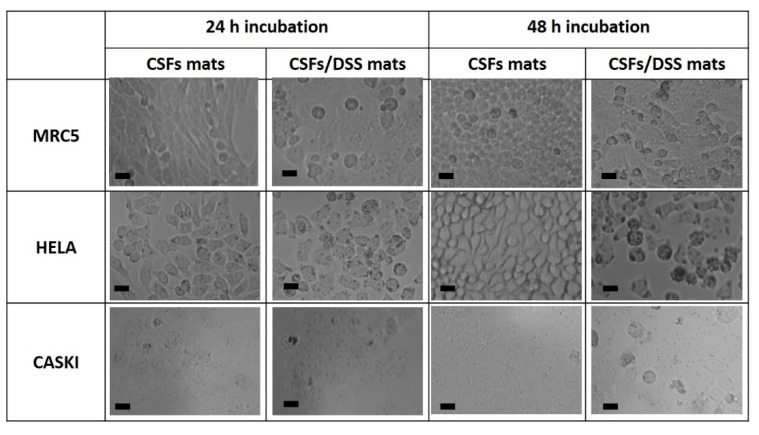
Microscopy images of cell line morphology. The scale bar corresponds to 20 µm.

**Figure 8 materials-13-03580-f008:**
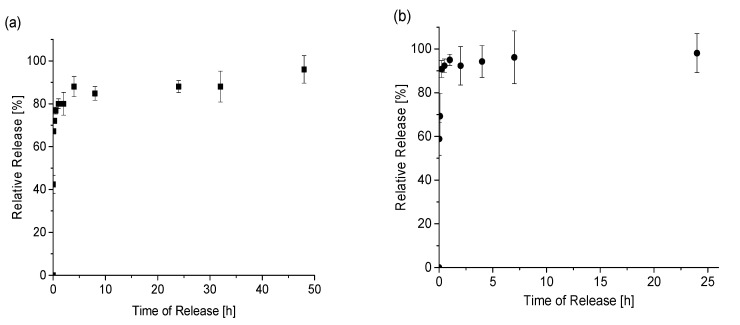
The relative release profile of DSS loaded into the electrospun CSF fibers with 15% *w*/*w* of DSS (**a**) and 30% *w*/*w* of DSS (**b**). Relative release was measured in triplicate from two independent release experiments.

**Figure 2 materials-13-03580-f002:**
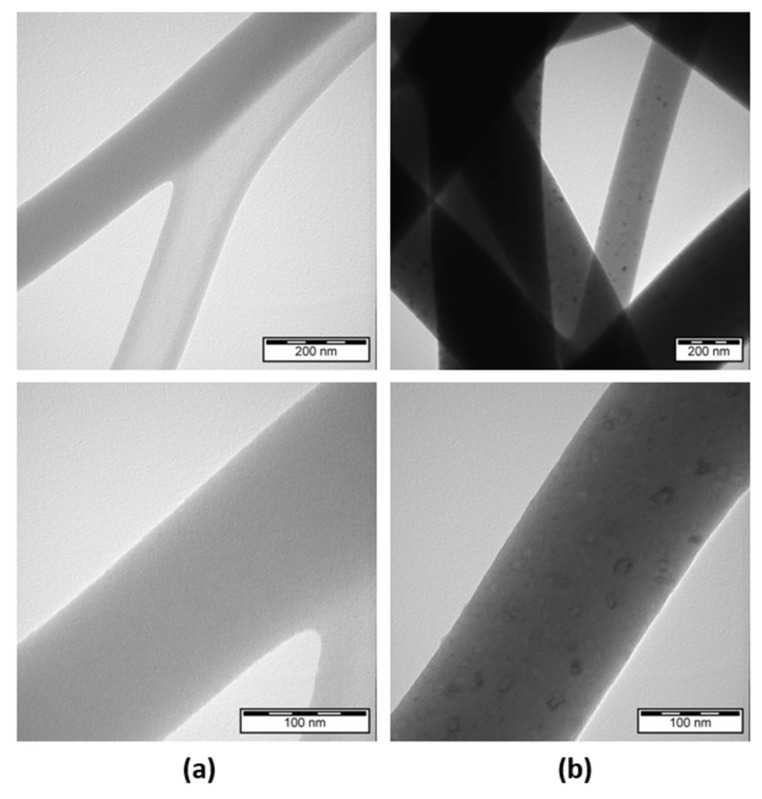
TEM images of electrospun silk fibroin composites with two different magnifications of 200 nm and 100 nm: (**a**) SFs; (**b**) CSFs/15%DSS.

**Table 1 materials-13-03580-t001:** Content of solutions prepared for electrospinning.

	Silk Fibroin (SFs)	DSS	Casein	HFIP
Silk fibroin (SFs)_	4 g	-	-	50 mL
Silk fibroin + DSS (SFs/15% or 30%DSS)	4 g	15% or 30%	-	50 mL
Silk fibroin + casein (CSFs)	4 g	-	5%	50 mL
Silk fibroin + casein + DSS (CSFs/15% or 30%DSS)	4 g	15% or 30%	5%	50 mL

**Table 2 materials-13-03580-t002:** The average diameter of individual electrospun fibrous mats A–F (see Figure 1) and their contact angles. All values are expressed as a mean ± SD.

Sample	Average Diameter [nm]	Contact Angle [°]
A (SFs)	162 ± 97	45 ± 1
B (CSFs)	301 ± 289	69 ± 4
C (SFs/15%DSS)	185 ± 95	41 ± 2
D (SFs/30%DSS)	238 ± 126	37 ± 3
E (CSFs/15%DSS)	199 ± 117	50 ± 3
F (CSFs/30%DSS)	191 ± 137	43 ± 3

Silk: 8% *w*/*v*; casein: 5% *w*/*v*; DSS: 15% or 30% *w*/*w* resp.

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
