# Peer review of "Diclofenac Embedded in Silk Fibroin Fibers as a Drug Delivery System"

_materials, 2020, doi:10.3390/ma13163580_

Round 1

Reviewer 1 Report

Although this paper reports quite interesting results on silk fibroin electrospun fibers, there is a very important issue, which the authors have to address. Diclofenac is not a cancer drug aimed at destroying or suppressing proliferation of malignant cells. Therefore, the choice of the drug is misleading. I do not see any real use or practical application of this material. What is the purpose of encapsulating a drug like diclofenac in such mats? What will be the practical use? This is unclear. The authors have to clearly explain that diclofenac is not an anticancer drug, and downplay all associated speculations. Other issues:

  • SEM is not capable to determine smoothness of roughness of any surfaces. SEM images presented are of very low resolution, please replace.
  • TEM images are not representative, please provide larger area images of both types of material
  • Is there any data based on EPR study? I failed to see any graph, etc.
  • TGA experimental details need to be presented
  • Did the authors obtain any microscopy images of the cells after cultivation? Was morphology affected?
  • Literature review needs to be improved. Most of the sources are outdated, with just two citations from 2017. There have been numerous studies since 2017, which have to be credited. Several related recent papers are recommended for citation:

Supranee Kaewpirom and Siridech Boonsang,  RSC Adv., 2020, 10, 15913-15923

Reizabal  et al, ACS Appl. Mater. Interfaces 2019, 11, 33, 30197–30206

Cho et al, ACS Appl. Nano Mater. 2018, 1, 10, 5441–5450

Atrian et al, Applied Clay Science, 2019, 17415, 90-99

Reference #15 is a PhD thesis, which I could not find, can it be replaced with a journal or book publication?

English is generally OK, but typos should have been checked.

Author Response

Response to Reviewer 1 Comments

We are indebted for the opinion of the Reviewer regarding our manuscript. In replying to specific comments, we would like to state that:

Point 1:

Although this paper reports quite interesting results on silk fibroin electrospun fibers, there is a very important issue, which the authors have to address. Diclofenac is not a cancer drug aimed at destroying or suppressing proliferation of malignant cells. Therefore, the choice of the drug is misleading. I do not see any real use or practical application of this material. What is the purpose of encapsulating a drug like diclofenac in such mats? What will be the practical use? This is unclear. The authors have to clearly explain that diclofenac is not an anticancer drug, and downplay all associated speculations.

Response 1:

The reviewer is wright about the fact that the diclofenac is not a typical anticancer drug. This drug was chosen as a model drug to study the SFs mats as delivery systems as is also written in the title of this manuscript. However, some anticancer properties of anti-inflammatory drugs were already published as reviewed in the introduction and therefore we decided to use the cancer cell lines as a proof of principle for release of the drug, which is still functional. As shown in Fig. 5, the released diclofenac decreased the cell viability of cancer cell lines proving that it was released and functional. We believe that the electrospun matts can be used also for other drugs as a delivery system (see line 85). The possible use of these materials in the future was added to the Conclusions (line 391).

Point 2:

SEM is not capable to determine smoothness of roughness of any surfaces. SEM images presented are of very low resolution, please replace.

Response 2:

The SEM micrographs of prepared material was replaced by micrographs with higher resolution. To be sure that the data given in Table 1, which are directly linked to the SEM images, are correct, we measured again the diameters of the fibers and we corrected them in Table 1. Despite minor changes in fiber diameters, the claims in the discussion remain valid (see Figure 1 and Table 1, line 224 and 236).

Point 3:

TEM images are not representative, please provide larger area images of both types of material.

Response 3:

We did not give larger areas because the fibers overlap and the informative value is lost. According to the comment of the Reviewer we exchanged one picture in the manuscript for the demonstration (see Figure 2b).

Point 4:

Is there any data based on EPR study? I failed to see any graph, etc.

Response 4:

The EPR graph has been added to the manuscript (see Figure 5, line 315).

Point 5:

TGA experimental details need to be presented.

Response 5:

The experimental details have been added to the manuscript (see line 156-159).

Point 6:

Did the authors obtain any microscopy images of the cells after cultivation? Was morphology affected?

Response 6:

Yes, we have obtained the microscopy images. New Figure 7 with images is included in the manuscript. The text in the manuscript was accordingly modified. The morphology of cells in the presence of CSFs mats did not change. In the presence of DSS loaded CSFs necrotic and dead cells are presented, the morphology of these cells is changed, they are of round shape, swollen with the signs of nucleus lysis (see Figure 7, line 351).

Point 7:

Literature review needs to be improved. Most of the sources are outdated, with just two citations from 2017. There have been numerous studies since 2017, which have to be credited. Several related recent papers are recommended for citation.

Response 7:

Thank you for your recommendation. References were updated.

Point 8:

Reference #15 is a PhD thesis, which I could not find, can it be replaced with a journal or book publication?

Response 8:

According to the comment of the Reviewer reference 15 was changed (see reference 19, line 452).

Reviewer 2 Report

Dear authors,

In the conclusions’ section I recommend to include a phrase with future perspectives of such materials. 367 Correct grammar: In comparison to the fibroblast cells line was investigated that the growth of both cancer lines was significantly inhibited after 48 h.

The references are old- only two references from 2017, the others are older. I recommend an update of the information and the references used.

Examples:

On diclofenac and cancer: https://pubmed.ncbi.nlm.nih.gov/?term=cancer+diclofenac&sort=date

On drug delivery systems in cancer: https://pubmed.ncbi.nlm.nih.gov/?term=nanosystems+cancer&sort=date

Author Response

Response to Reviewer 2 Comments

We are indebted for the opinion of the Reviewer regarding our manuscript. In replying to specific comments, we would like to state that:

Point 1:

In the conclusions’ section I recommend to include a phrase with future perspectives of such materials.

Response 1:

The possible use of these materials in the future was added to the Conclusions (line 391).

Point 2:

367 Correct grammar: In comparison to the fibroblast cells line was investigated that the growth of both cancer lines was significantly inhibited after 48 h.

Response 2:

According to the comment of the Reviewer sentence was corrected (line 388-389).

Point 3:

The references are old- only two references from 2017, the others are older. I recommend an update of the information and the references used.

Examples:

On diclofenac and cancer: https://pubmed.ncbi.nlm.nih.gov/?term=cancer+diclofenac&sort=date

On drug delivery systems in cancer: https://pubmed.ncbi.nlm.nih.gov/?term=nanosystems+cancer&sort=date

Response 3:

According to the comment of the Reviewer most references have been replaced or updated by newer ones and are marked in the manuscript.

Reviewer 3 Report

Reviewed manuscript is well written. The work was well conducted, and the results were adequately presented and discussed. In my humble opinion, this manuscript may be published after minor corrections.

Specific comments:

  • Is the thickness of all prepared nanofibers layers, which were used for contact angle measurements, was the same? I doubt the correctness of contact angle measurements. Please provide some photos of obtained results. In my opinion, water during measurement can reach aluminum foil, which was used for collection of nanofibers. In such case the provided results can not display real contact angle values.
  • TEM results, presented in figure 2, were provided as evidence of casein and diclofenac loading into nanofibers. In my opinion, it is too early to say for sure. The deference in structures of SFs and CSFs/15%DSS are seen well, but it is not possible to claim the nature of black spots in CSFs/15%DSS structure just from TEM results.
  • The citation is needed for statement, which was presented in text (lines 303-305).
  • In figure 5 results of SFs biocompatibility are presented. Did the same experiments was done for CSFs mats?

Author Response

Response to Reviewer 3 Comments

We are indebted for the opinion of the Reviewer regarding our manuscript. In replying to specific comments, we would like to state that:

Point 1:

Is the thickness of all prepared nanofibers layers, which were used for contact angle measurements, was the same? I doubt the correctness of contact angle measurements. Please provide some photos of obtained results. In my opinion, water during measurement can reach aluminum foil, which was used for collection of nanofibers. In such case the provided results can not display real contact angle values.

Response 1:

For the contact angle measurements, free-standing electrospun scaffolds were used, so the aluminum foil can have any impact for the results of the measurement. The thickness of the measured nanofibers layers was 25 ± 1 mm (see line 161-166).

For the illustration (attached document) you can see the sample from measurement: 8% Silk in HFIP with 15% of diclofenac sodium salt from wt.% of silk.

α = 41.165 ± 2.418o

We were inspired by available literature and the wettability is commonly assessed by contact angle measurement.

Yao, H. Wang, R. Wang, Y. Chai. Preparation and characterization of homogenous and enhanced casein protein-based composite films via incorporating cellulose microgel. Scientific Reports 2019, 9, 1221-1232. https://doi.org/10.1038/s41598-018-37848-1

Minaei, S.A.H. Ravandi, S.M. Hejazi, F. Alihosseini. The fabrication and characterization of casein/PEO nanofibrous yarn via electrospinning. E-Polymers 2019, 19(1), 154-167. https://doi.org/10.1515/epoly-2019-0017

Wang, M. Lin, Q. Xie, H. Sun, Y. Huang, D.D. Zhang, Z. Yu, X. Bi, J. Chen, J. Wang, W. Shi, P. Gu, X. Fan. Electrospun silk fibroin/poly(lactide-co-ε-caprolactone) nanofibrous scaffolds for bone regeneration. Int. J. Nanomedicine 2016, 11, 1483-1500. 10.2147/IJN.S97445

Nazari, A.H. Tabasi, M. Hajiabbas, M.S. Bani, M. Nazari, V.P. Mahabadi, I. Rad, M. Kehtari, S.H.A. Tafti, M. Soleimani. J. Cell. Biochem. 2020, 121(4), 2981-2993.  10.1002/jcb.29553

Point 2:

TEM results, presented in figure 2, were provided as evidence of casein and diclofenac loading into nanofibers. In my opinion, it is too early to say for sure. The deference in structures of SFs and CSFs/15%DSS are seen well, but it is not possible to claim the nature of black spots in CSFs/15%DSS structure just from TEM results.

Response 2:

We agree. With the results of TEM, we wanted to demonstrate changes in the structure after the addition of DSS and casein.

Point 3:

The citation is needed for statement, which was presented in text (lines 303-305).

Response 3:

In our opinion citation is not needed because it is our contribution to the discussion following from our experiments (see line 311-313).

The EPR graph has been added to the manuscript (see Figure 5, line 315).

Point 4:

In figure 5 results of SFs biocompatibility are presented. Did the same experiments was done for CSFs mats?

Response 4:

We apologize for this mistake, but the results are valid for CSF mats, the SF mats were not studied. Figure 5 was corrected (see Figure 6, line 327).

Round 2

Reviewer 1 Report

The authors did a good work in revising the MS, however there are still certain issues to be addressed:

  1. This statement "Scanning electron microscopy images show that silk nanofibers obtained from 8% w/v 1,1,1,3,3,3-hexafluoro-2-propanol solution are smooth and bead-free with an average diameter in the range of 171-254 nm. " is not correct. SEM does not provide ANY information regarding actual smoothness or roughness of any surface. Smoothness can be evaluated only using a certain profilometry technique (i.e. AFM would be most appropriate in this case). Therefore, the authors have to change the term used, for example they can say that the fibres' surfaces lack any visible features, or something similar. But using any reference to smoothness should be avoided.  
  2. The authors should cite the following papers as highly relevant to the study reported.

Supranee Kaewpirom and Siridech Boonsang,  RSC Adv., 2020, 10, 15913-15923

Reizabal  et al, ACS Appl. Mater. Interfaces 2019, 11, 33, 30197–30206

Cho et al, ACS Appl. Nano Mater. 2018, 1, 10, 5441–5450

Author Response

Response to Reviewer 1 Comments

We are indebted for the opinion of the Reviewer regarding our manuscript. In replying to specific comments, we would like to state that:

Point 1:

This statement "Scanning electron microscopy images show that silk nanofibers obtained from 8% w/v 1,1,1,3,3,3-hexafluoro-2-propanol solution are smooth and bead-free with an average diameter in the range of 171-254 nm. " is not correct. SEM does not provide ANY information regarding actual smoothness or roughness of any surface. Smoothness can be evaluated only using a certain profilometry technique (i.e. AFM would be most appropriate in this case). Therefore, the authors have to change the term used, for example they can say that the fibres' surfaces lack any visible features, or something similar. But using any reference to smoothness should be avoided.  

Response 1:

The sentence was modified, according to the reviewer's request: “Scanning electron microscopy images show that silk nanofibers obtained from 8% w/v 1,1,1,3,3,3-hexafluoro-2-propanol solution  lack any visible features and have average diameter in the range of 171-254 nm.” (line 22-24).

Point 2:

The authors should cite the following papers as highly relevant to the study reported.

Supranee Kaewpirom and Siridech Boonsang,  RSC Adv., 2020, 10, 15913-15923

Reizabal  et al, ACS Appl. Mater. Interfaces 2019, 11, 33, 30197–30206

Cho et al, ACS Appl. Nano Mater. 2018, 1, 10, 5441–5450

Response 2:

Suggested references were used in the manuscript (see reference 18-20, line 64).